# Super-Long SERS Active Single Silver Nanowires for Molecular Imaging in 2D and 3D Cell Culture Models

**DOI:** 10.3390/bios12100875

**Published:** 2022-10-15

**Authors:** Xiao-Tong Pan, Xuan-Ye Yang, Tian-Qi Mao, Kang Liu, Zao-Zao Chen, Li-Na Ji, De-Chen Jiang, Kang Wang, Zhong-Ze Gu, Xing-Hua Xia

**Affiliations:** 1State Key Laboratory of Analytical Chemistry for Life Science, School of Chemistry and Chemical Engineering, Nanjing University, Nanjing 210023, China; 2Institute of Theoretical and Computational Chemistry, Key Laboratory of Mesoscopic Chemistry of the Ministry of Education (MOE), School of Chemistry and Chemical Engineering, Nanjing University, Nanjing 210023, China; 3State Key Laboratory of Bioelectronics, School of Biological Science and Medical Engineering, Southeast University, Nanjing 210096, China; 4State Key Laboratory of Pharmaceutical Biotechnology, School of Life Sciences, Nanjing University, Nanjing 210023, China

**Keywords:** Ag nanowire, cell culture model, SERS, machine learning, spatial resolution

## Abstract

Establishing a systematic molecular information analysis strategy for cell culture models is of great significance for drug development and tissue engineering technologies. Here, we fabricated single silver nanowires with high surface-enhanced Raman scattering activity to extract SERS spectra in situ from two-dimensional (2D) and three-dimensional (3D) cell culture models. The silver nanowires were super long, flexible and thin enough to penetrate through multiple cells. A single silver nanowire was used in combination with a four-dimensional microcontroller as a cell endoscope for spectrally analyzing the components in cell culture models. Then, we adopted a machine learning algorithm to analyze the obtained spectra. Our results show that the abundance of proteins differs significantly between the 2D and 3D models, and that nucleic acid-rich and protein-rich regions can be distinguished with satisfactory accuracy.

## 1. Introduction

Cell culture models provide an in vitro platform for basic and clinical research. A two-dimensional (2D) cell culture model is traditionally used in the study of cellular characteristics and drug screening, owing to its simplicity, reproducibility and low cost. In a 2D model, cells are usually seeded on a flat surface, and grow fast into a monolayer; however, it fails to accurately reflect the three-dimensional (3D) architecture of living tissues [1]. In contrast to a 2D model, 3D cell culture model better recapitulates the crucial features of complex in vivo environment. Similar to solid tumors, 3D multicellular tumor spheroids can provide an in vitro tumor microenvironment, which is valuable for drug screening [2,3,4]. Three-dimensional organoids that are formed by self-organized stem cells can recreate phenotypic and functional traits of the original biological specimens [5,6]. The 3D model has been widely used in drug development, tissue regeneration and transplantation, but the lack of proper analysis and evaluation methods limits its large-scale application [7,8]. Compared with other established analysis strategies, such as compartmentalization [9], genomics [10], transcriptomics [11] and proteomics [12,13], imaging is usually preferred for analysis of 3D model because it enables in situ visualization of cells and retains spatial molecular information [14,15,16]. Traditional imaging techniques, such as transmission electron microscopy (TEM) and scanning electron microscopy (SEM), have been used to analyze multicellular tumor spheroids. However, these techniques cannot be performed on live cells. Furthermore, the imaging depth is also a limitation. The 3D samples (dozens to thousands of microns in diameter) are often fixed, embedded and sectioned before imaging, thus possibly introducing artefacts and deformities. Fluorescence imaging has also been used to provide molecular information in a 3D model, but its application is limited by imaging depth and the requirement of sample labeling [17,18].

Superior to fluorescence, TEM and SEM, Raman spectrum can provide spectral fingerprints as well as spatial distribution of endogenous biomolecules [19,20]. The unique Raman spectrum of each molecule reflects the vibration energy level of its chemical bonds, which guarantees the multivariate analysis of label-free biomolecules including lipids, nucleic acids and proteins [21,22]. When a rough plasmonic metal (usually Ag or Au) is used as a substrate for Raman detection, so-called surface-enhanced Raman scattering (SERS) can occur, which can greatly increase the sensitivity of Raman spectroscopy and the imaging depth along the Z-axis [23,24]. However, planar substrates can only enhance the molecular spectra on the surface rather than the interior of the 3D model [25,26]. Metal nanoparticles can enter cells through endocytosis and serve as SERS substrates, but can hardly be attached to target organelles such as the nucleus [19,27]. Consequently, fabricating a Raman-enhanced substrate that is applicable in a 3D model can help us understand the bio-process of ex vivo tissue and develop a tumor model and organoid chips.

For the 3D model, SERS substrates that can be inserted through multiple cells on demand are desirable because the spectral signals at multiple locations can be detected and compared in parallel. Although precisely manipulated glass pipettes [28,29] and metal tips [30,31] have been inserted into single cells with minimal damage, such conical shaped probes can hardly be used for a 3D model because their damage to cells becomes serious during their invasion from outer to inner layers. Attaching one-dimensional materials such as carbon nanotubes [32,33] or gold nanorods [34,35] to a probe tip can keep to minimal cell damage in multicellular detection. An ultra-long one-dimensional SERS substrate with a submicron diameter is ideal for acquiring molecular information from a 3D model.

Besides ideal SERS substrates, effective spectral analysis techniques are also important for the establishment of a systematic analysis strategy. Considering the weak signals of analytes and overlapping features of multivariate information in biological systems, it is necessary to introduce systematic computational ways. Principal component analysis (PCA) is a multivariate analysis method that can reduce the dimensionality of original data to highlight the core information. Thus, PCA is often used to capture key features from noisy Raman spectra. However, challenges remain in correlating the acquired SERS spectra with the intracellular dominant molecular species [36,37].

Both Random Forest and K-Means++ are efficient algorithms in the field of machine learning for solving classification problems [38,39]. K-Means++ is an unsupervised clustering algorithm that can cluster unlabeled data, whereas Random Forest is a supervised algorithm with very high accuracy and efficiency on large data set. Therefore, Random Forest can learn the labels of K-Means++ and rank the importance of variables, while K-Means++ can classify data according to the importance of variables. The combination of Random Forest and K-Means++ may identify the dominant molecular species in the spectra. As far as we known, the application of the combined algorithms in spectral analysis has not been reported yet.

We have developed an approach for in situ growth of single silver nanowires (AgNWs) with ultrahigh aspect ratio [40]. Here, we modified the method and fabricated a single AgNW with high SERS-activity on a nanoelectrode to acquire biomolecular information in 2D and 3D models. The fabricated AgNW was mounted on a four-dimensional (4D) microcontroller acting as a cell endoscopy. The SERS spectra of multiple cells were collected along the AgNW in both 2D and 3D models. Then, we adopted PCA and machine learning algorithms for the spectrum analysis. Our results indicated that nucleic acid-rich and protein-rich regions in 2D and 3D models could be distinguished with satisfactory accuracy.

## 2. Experimental Section

### 2.1. Synthesis of Single AgNWs

Based on our previously reported method with modification [40], a quartz capillary with an inner diameter of 100 nm was first fabricated by a CO_2_-laser-based pipet puller (P-2000, Sutter Instrument Co.) from quartz (O.D. 1.0 mm, I.D. 0.70 mm, Sutter, QF100-70-10). The parameters are shown below:

HEAT = 700, FIL = 3, VEL = 40, DEL = 175, PUL = 190.

A carbon nanoelectrode was then made by depositing a layer of carbon on the inner surface of the quartz capillary. Butane and argon were used as the carbon source and protector in the chemical vapor deposition (CVD), respectively.

The carbon nanoelectrode was immersed in 10 mM AgNO_3_ solution containing 2.5 μM trisodium citrate. An Ag/AgCl wire was used as the auxiliary electrode. A chrono potentiometry was applied on the carbon nanoelectrode by a potentiostat (CHI 660E, CH Instruments). A constant reduction current of 30 nA was used to synthesize an Ag ball for about 5 s. Subsequently, a reduction current of 2 nA was applied to grow a single AgNW. When the length of the AgNW reached about 150 μm, the electrochemical synthesis was manually terminated. All the electrochemical syntheses were performed at room temperature.

### 2.2. SERS Measurements on Single AgNWs

Raman spectra were obtained on a Renishaw InVia Reflex confocal microscope equipped with a high-resolution grating (1800 grooves/cm). An He-Ne laser (λ = 633 nm, laser power at spot = 3 mW) and a long working distance lens (50×, N.A. = 0.5) were used to provide a focus of around 1 μm in diameter. The spectrometer was calibrated by the Raman band of a silicon wafer at 520 cm^−1^. Compared with exciting lasers with other wavelengths (488 and 785 nm), the 633 nm laser produces stronger plasmonic resonance on the AgNW. Therefore, we chose 633 nm as the exciting laser.

For the measurement of Raman enhancement factor (EF), AgNW was immersed in 10^−9^ M Rhodamine 6G (R6G) solution. The integration time was 20 s and the laser intensity was 100%. For Raman mapping, AgNW was first immersed in 4-mercaptobenzonitrile (4-MBN) solution (100 mM in DMF) for 12 h. Then, the Raman signal was measured in air with an integration time of 2 s (10% laser intensity) and scanning steps of 1 μm.

For SERS measurement in 2D and 3D models, a single AgNW fabricated at the tip of the carbon nanoelectrode was first inserted into cells with the help of a 4D microcontroller (RMA-AWC03, Rayme, China). Subsequently, SERS detection was performed in situ with an integration time of 2 s, scanning steps of 1 μm and 100% laser intensity with the cells in phosphate-buffered saline (PBS, pH = 7.4).

### 2.3. Cell Culture

MCF-7 breast cancer cells were cultured at 37 °C in a humidified incubator with 5% CO_2_ for 40 h. We chose densely distributed monolayer cells as the 2D model to conduct experiments. The medium used was Dulbecco’s modified eagle medium supplemented with 10% fetal bovine serum and 1% penicillin–streptomycin. Glass bottom cell culture dishes were used in SERS detection to avoid signal interference.

The 3D model was constructed by the method reported previously [7]. Briefly, MCF-7 breast cancer cells were seeded on 96-well non-adhesive U-bottom cell culture plates at 10^5^ cells/well, and solid spheroid formed after four days of culture. Unlike the reported method, no Matrigel was added. The culture medium was the same as that used for the 2D model. Before SERS measurements, we transferred the tumor spheroid to glass bottom cell culture dishes for 12 h to ensure its attachment to the dish bottom.

### 2.4. Principal Component Analysis

PCA is usually adopted to reduce the dimensionality of data. The Raman spectra were collected in wavenumbers ranging from 717 to 1827 cm^−1^, showing 1015 features. Python codes were used in PCA to identify the principal components (PCs) and proportions of each PC.

The K-nearest neighbor (KNN) algorithm is one of the simplest and commonly used classification algorithms. The first principal component (PC1) and the second principal component (PC2) are applied to distinguish different Raman spectra. We defined the root mean square differences of proportions in PC1 and PC2 among Raman spectra as the distances in KNN. The ratio of correctly distinguished spectra over total test spectra was used as a score to evaluate the machine learning model (at K = 11).

### 2.5. Random Forest and K-Means++ Clustering Algorithm

K-means++ implementation in a python scikit-learn library was used to explore the clusters of the unlabeled spectra. The input of K-means++ was the Raman data ranging from 717 to 1827 cm^−1^, and the cluster number ranging from 1 to 5. By calculating the silhouette score of each number, 2 was found to be the most suitable cluster number for our data. Random Forest implemented by the python scikit-learn library was then used to calculate the importance of each wavenumber, and several important spectral bands were identified. Subsequently, two important spectral bands were chosen as the input for K-means++, which was used to classify the Raman spectra. Meanwhile, the normalized distance between the cluster center and each spectrum within this cluster was calculated to illustrate the intensity of the spectrum.

## 3. Results and Discussion

### 3.1. Synthesis and Characterization of Single AgNWs

To extract the molecular information from the cell culture model, we fabricated an ultra-long single AgNW at the tip of a carbon nanoelectrode based on our previously reported method with modification [40]. The experimental setup is shown in Figure 1a. The carbon nanoelectrode was ~100 nm in diameter (Appendix A) and connected with a potentiostat by a silver wire. When a reduction current was applied to the carbon nanoelectrode, Ag^+^ was reduced to Ag and formed a single AgNW in situ. For the synthesis of anisotropic structure, citrate was employed as capping agent in our case. Citrate can be selectively adsorbed on the Ag(100) surface resulting in easier atomic deposition on the Ag(111) surface (Figure 1b). The balance between citrate adsorption and Ag deposition on the side surface determines the surface roughness of the AgNW (Appendix A). As a shape directing agent, the citrate tends to be absorbed onto the AgNW and keeps the surface smooth. In order to enhance the Raman activity of AgNWs, we applied a relatively large reduction current of 2 nA. Therefore, the speed of Ag deposition was relatively faster than that of citrate adsorption, resulting in an AgNW with rough surface (Figure 1c). The diameter of the electrochemically synthesized single AgNWs was 300 ± 50 nm (Figure 1c), which is thin enough to keep the cells alive. The length of the AgNWs was controlled between 100–200 μm by adjusting reduction time (Figure 1d), so they are long enough to penetrate 5~15 cells. These features make AgNWs appropriate SERS substrates for 2D and 3D models. It is worth noting that an Ag ball was generated by electrochemical reduction prior to the growth of AgNW (Figure 1e). Usually, an Ag ball smaller than 2 μm will not affect the electrochemical synthesis of AgNW and can act as a solid connection between the AgNW and the nanopipette, which facilitates the penetration of AgNWs through cells. When we applied a fast vibration perpendicular to the direction of the AgNW extension, the Ag ball would be easily detached from the nanoelectrode, allowing convenient in situ SERS measurement (Appendix A).

As indicated in Figure 1f, the rough-surface AgNW showed super-high SERS activity, which can be attributed to the presence of numerous “hot spots”. The Raman enhancement factor of AgNW was calculated to be 2.53 × 10^5^ using R6G as a probe molecule (Appendix A). In order to demonstrate the distribution of hot spots along AgNW, SERS linear mapping was also implemented using 4-MBN as a probe molecule. As shown in Figure 1g, a strong SERS signal of CN- group at 2227 cm^−1^ could be detected along AgNW. The band area of the CN-group on AgNW gave a relative standard deviation around 4.22% (34 data points, Appendix A). The high SERS activity and even distribution of hot spots revealed that AgNW could be a powerful tool for extracting molecular information from the microenvironment.

### 3.2. Raman Spectra Acquisition in 2D and 3D Models

As shown in Figure 2a, the nanoelectrode with an AgNW was manipulated by a 4D microcontroller during the insertion process. Because the diameter of AgNW varies very little from one end to the other, its damage to the outer-layer cells might be as minimal as that to the inner-layer cells. Simultaneously, the highly flexible AgNW could easily pass through multiple cells without breaking. As shown in Figure 2b and Appendix A, AgNWs could pierce several consecutive cells in the 2D monolayer cell model along the dish bottom. In the spherical 3D model, AgNWs could also be inserted into cells in different depths (Figure 2c and Appendix A). Therefore, we are able to extract and compare SERS spectra of different cells in parallel to analyze the biological properties of different models.

For SERS mapping, the incident laser scanned on multiple spots along AgNW, which provided spatial distribution information of molecules in cell culture models. Figure 2d,e show the typical SERS contour graphs obtained from 2D and 3D models, respectively. More SERS spectra are shown in Appendix A. For each model, 600 to 800 spectra were collected using more than 10 AgNWs. SERS spectra shows reliable reproducibility on different AgNWs. As shown in Figure 2f, the average spectra of 2D and 3D models demonstrated remarkable similarity in several strong bands as expected, because both models were formed by MCF-7 cells. The tentative assignments of those strong bands are listed in Table 1. In both 2D and 3D models, most of the bands could be assigned to the vibrations of lipids, proteins and nucleic acid [19,20,25]. However, it is difficult to analyze the spectra in depth by visual observation of the presence/absence of peaks or ratios between different peaks. Therefore, we introduced a computational way to identify biomolecules in spatial resolution.

### 3.3. Multivariate Analysis of Raman Spectra

We used PCA to capture key features in cell culture models. As reported, PCA provided orthogonal PCs from the eigenvectors of the covariance matrix for Raman spectra and proportions of PCs from eigenvalues [41]. PCs were arranged in descending order of proportions. Projecting the entire data into the two-dimensional linear space generated by PC1 and PC2 made it easier to find key features from Raman spectra. As shown in Figure 3a, we applied PCA to the Raman spectra of the 2D and 3D models, and projected the results into a two-dimensional PCA score space. We used 95% confidence ellipses to represent the range of projected points corresponding to each model. The two ellipses did not overlap, indicating that PCA can clearly distinguish between 2D and 3D models.

We further identified the key feature bands corresponding to each model from the loading plot of PCs (Figure 3b). In principle, PCA produces different weights on each Raman wavelength to find the direction of the maximum variance of the data. Therefore, the Raman shifts with high-level loading values (positive or negative) can be used as the key feature bands to distinguish the spectra of 2D and 3D models. As shown in Figure 3a, only the 2D model was projected into the fourth quadrant (PC1 > 0 and PC2 < 0), while the 3D model was projected into the second quadrant (PC1 < 0 and PC2 > 0). The above difference indicated that the Raman shifts that resulted in positive values for PC1 and negative values for PC2 were the key features of the 2D model (grey bands shown in Figure 3b). Meanwhile, the Raman shifts that resulted in negative values for PC1 and positive values for PC2 were the key features of the 3D model (red bands). Table 2 lists the characteristic Raman bands corresponding to the 2D and 3D models read in Figure 3b and their tentative assignments. As can be seen in Table 2, the characteristic Raman bands of lipids and nucleic acid appear in both 2D and 3D models. However, all the significant characteristic protein bands appear in the 3D model alone, indicating a higher abundance of proteins in 3D model. It has been reported that 3D spheroids displayed an increased extracellular matrix (ECM) of proteins [42] such as collagen and fibronectin, in a similar manner to in vivo tumors [43,44]. Our result falls in line with the reports, suggesting that the characteristic protein bands at 851, 1002, 1246 and 1662 cm^−1^ were possibly originated from the increased ECM proteins in 3D model. Therefore, the characteristic Raman bands of proteins could possibly be used as one of the criteria for evaluating the tumor model or other biochips.

To validate the above key features for distinguishing between 3D and 2D models, we built a simple machine learning model using the KNN algorithm. We randomly selected 80% of the Raman spectra of the two models as the training set and the remaining 20% as the test set. As shown in Figure 3c, all 246 spectra in the test sets perfectly overlapped with the corresponding train sets. The machine learning model received a score of 1, which showed that all the test spectra were correctly discriminated. The perfect score confirmed the above conclusion that PCA could find key features of the 3D model and distinguish it from the 2D model.

In addition to PCA characterization of the molecular species in cell culture models, we also explored the distribution of molecules along single AgNWs. Conventional spatial resolution was based on observing the absolute intensity of the characteristic Raman band in SERS mapping, which puts forward high requirements for the signal-to-noise ratio and stability of the data [20]. Therefore, it is difficult to achieve effective spatial resolution of analytes from complex samples.

We introduced an algorithm combining Random Forest with K-means++ for further identifying the dominant cellular components distributed along single AgNW. A schematic diagram of the algorithm is shown in Appendix A. Using the 2D model as an example, we used Random Forest to learn the K-means++ labels of the full spectra data. The importance distribution map of the Raman shift was obtained using Random Forest, as shown in Figure 4a. Four important bands with large distribution differences in the 2D model could be read out. The bands located around 1655 and 782 cm^−1^ showed the highest intensity, which revealed the most important Raman bands. Combined with the average SERS spectrum shown in Figure 4b, the bands at 1655 and 782 cm^−1^ could be assigned to amide I of proteins and nucleic acid, respectively (Figure 4c). Based on the Random Forests algorithm, Figure 4a,c suggested that the SERS spectra collected along the AgNW in 2D model could reflect the abundance of proteins and nucleic acid in different subcellular locations. It is possible to correlate the SERS spectral information with the subcellular distribution of endogenous biomolecules, considering the relatively high abundance of proteins in the cytoplasm and nucleic acids in the cell nucleus.

Subsequently, the two most important spectral bands at 1655 and 782 cm^−1^ were chosen as the input for unsupervised clustering algorithm K-means++ to obtain the subcellular distribution information of proteins and nucleic acids. The K-means++ algorithm classified the spectra along AgNW into two categories: protein-rich regions (green) and nucleic acid-rich regions (red). In the algorithm, the normalized distance between the cluster center and each spectrum was used to indicate the intensity of the molecular signal. After the classification, we combined the molecular distribution information with the spatial position in the optical image to show the distribution of nucleic acids and proteins in the monolayer cells. As indicated in Figure 4d, there are more red dots in the cell nucleus area (red circle), showing that nucleic acids were detected (upper right). The co-existence of a few green dots suggested that nucleoprotein could also be identified. Meanwhile, the ECM (area between the two green curves) and the cytoplasm (green circle) overlapped well with the green dots, suggesting that higher abundance of proteins existed in both areas (bottom right). We also calculated the Silhouette Coefficient (SC) to evaluate the clustering performance of this algorithm. The SC is given by Equation (1):(1)$$\mathrm{SC}=\frac{1}{N}\overset{N}{\underset{i=1}{\sum }}\frac{b_{i}-a_{i}}{\underset{}{\max}\left(a_{i},b_{i}\right)}$$
where $a_{i}$ is the average distance of a sample from other samples in the same cluster, $b_{i}$ is the average distance between a sample and samples in other clusters. The SC value was calculated to be 0.1838 (>0) in our case, which proved a good clustering performance of our strategy.

Compared with the 2D model, the 3D model allows lower light transmittance. It is very difficult to observe the individual cells inside the 3D model as well as to acquire spatially resolved spectral information. We applied the above Random Forest-combined K-means++ algorithm to the 3D model. The corresponding Raman shift importance distribution map is shown in Appendix A. As shown in the Figure 4e, the optical microscope can only show the morphology of surface layer cells in the 3D model (yellow rectangle). Upon applying the above algorithm to the 3D model, the spectral data could be clearly divided into two groups (blue rectangle) representing the protein-rich regions and nucleic acid-rich regions, respectively. The analytical results showed that Random Forest-combined K-means++ algorithm was powerful for processing SERS spectra of inner cells in a 3D model where the cell boundaries could not be observed.

## 4. Conclusions

Applying a single AgNW as a Raman-enhanced substrate was an efficient strategy for obtaining molecular information from the SERS spectra of 2D and 3D models. The PCA result suggested that the different protein abundance in 2D and 3D models resulted in the differences between their spectra. The KNN algorithm could make a perfect distinction between the two cell culture models based on the PCA result. Our strategy suggests a new way for analyzing cell culture models using Raman spectroscopy. The strategy is promising to be further optimized by improving analyte specificity and tissue optical clearing technique detection.

## Figures and Tables

**Figure 1 biosensors-12-00875-f001:**
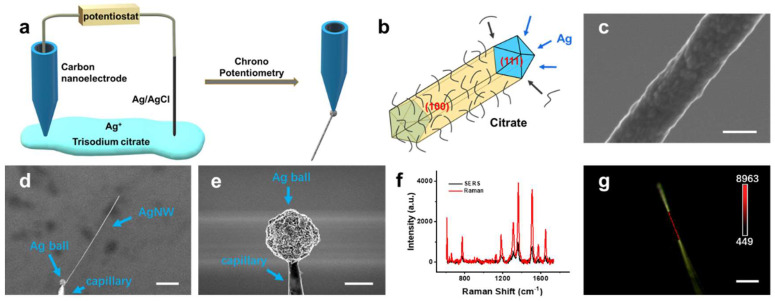
(**a**) Schematic diagram for the synthesis of single AgNWs by an electrochemical method. (**b**) Schematic diagram for the mechanism underlying the unidirectional growth of AgNWs. (**c**,**d**) SEM images of an AgNW. (**e**) SEM image of the Ag ball at the carbon nanoelectrode tip. (**f**) SERS spectrum (black) of 10^−9^ M R6G on an AgNW and Raman spectrum (red) of 10^−3^ M R6G. (**g**) SERS mapping of a single AgNW modified with 4-MBN. Scale bars: (**c**) 200 nm; (**d**) 20 μm; (**e**) 2 μm; (**g**) 20 μm.

**Figure 2 biosensors-12-00875-f002:**
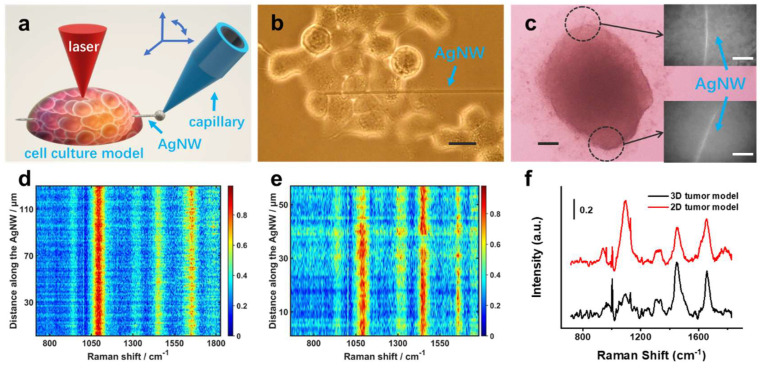
(**a**) Schematic diagram for inserting single AgNW into a cell culture model. (**b**,**c**) Optical images of a 2D model inserted with an AgNW (**b**) and a 3D model inserted with two AgNWs (**c**). (**d**,**e**) Contour graphs of Raman spectra along AgNW. (**d**) A 2D model with 130 spectra, and (**e**) a 3D model with 57 spectra.(**f**) Average spectra of 2D and 3D models. The number of spectra used for the average spectrum calculation were 270 (2D) and 200 (3D), respectively. Scale bars: (**b**) 20 μm; (**c**) 100 μm and (**c**) inset 30 μm.

**Figure 3 biosensors-12-00875-f003:**
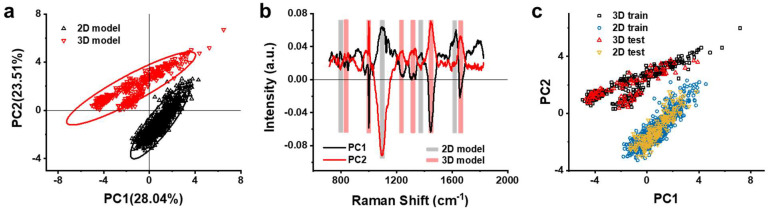
(**a**) Projection of Raman spectra onto a 2D PCA score space for the 2D and 3D models; 95% confidence ellipses enclosing the projected spectra as dots on the PCA score space are also shown. (**b**) Loading plot of the PC1 and PC2 for the spectra of 2D and 3D models; the black line represents the loading of PC1, and the red line represents the loading of PC2, the grey and red bands indicate the characteristic Raman shifts for the 2D and 3D models, respectively. (**c**) Results of the classification prediction for the PCA of the 2D and 3D models using the KNN algorithm.

**Figure 4 biosensors-12-00875-f004:**
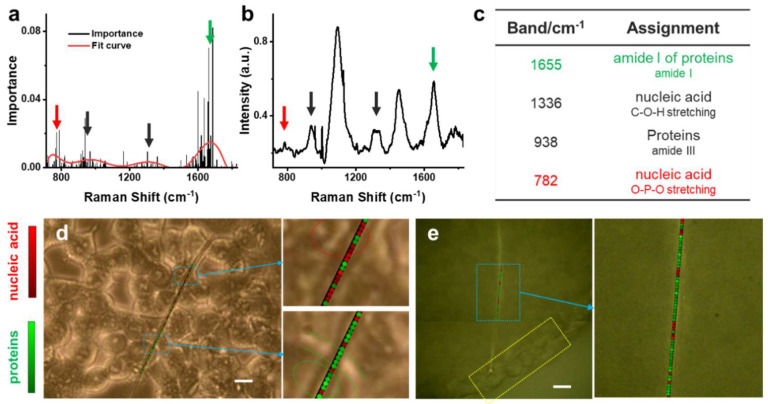
(**a**) Importance of Raman shift learned from 803 SERS spectra of 2D model using Random Forest algorithm. The red curve represents the fit curve of the importance. Arrows point out four important Raman bands. The red and green arrows point out the two most important bands, respectively. (**b**) Average spectrum of all the 803 SERS spectra obtained from 2D model. Arrows point out the four corresponding bands in (**a**). (**c**) Raman shifts and tentative assignments of the important Raman bands found by Random Forest. (**d**) Distribution of proteins and nucleic acids in 2D model using K-means++ algorithm. In the upper panel, the red circle indicates cell nucleus. In the bottom panel, the green circle indicates cytoplasm while the area between two green curves indicates the area of ECM. (**e**) Distribution of proteins and nucleic acids in 3D model using K-means++ algorithm. The optical microscope shows the morphology of outer layer cells (yellow rectangle) in 3D model. The inset (blue rectangle) represents the protein-rich regions and nucleic acid-rich regions in 3D model after the spectral data were analyzed by the machine learning algorithm. Scale bars in (**d**) and (**e**) 20 μm.

**Table 1 biosensors-12-00875-t001:** Tentative assignments of important bands in the Raman spectra of 2D and 3D models [19,20,25].

Bands (2D) [cm^−1^]	Bands (3D) [cm^−1^]	Tentative Assignment
1782	1785	Lipids, C=O stretching
1655	1657	Proteins, amide I
1449	1449	Lipids, -CH_2_- deformation
1306~1339	1306~1338	Lipids, proteins & nucleic acid
1128	1128	Proteins, C-N and C-C stretching
1097	1093	Phospholipids, C-C stretching
1003	1002	Amino acid, Phe
937	938	Proteins, amide III
782	782	Nucleic acid, O-P-O stretching

**Table 2 biosensors-12-00875-t002:** Raman shifts and tentative assignments of characteristic Raman bands found from PCA for 2D and 3D models [19,20,25].

Bands [cm^−1^]	2D Model	3D Model
1662		Proteins, amide I
1628	Lipids, C=C stretching	
1448		Lipids, -CH_2_- deformation
1377	Lipids, -CH_3_- deformation	
1336		Nucleic acid, C-O-H stretching
1246		Proteins, amide III
1096	Phospholipids, C-C stretching	
1002		Amino acid, Phe
851		Amino acid, Tyr
794	Nucleic acid, O-P-O stretching	

## Data Availability

Not applicable.

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
