# Peer review of "Super-Long SERS Active Single Silver Nanowires for Molecular Imaging in 2D and 3D Cell Culture Models"

_biosensors, 2022, doi:10.3390/bios12100875_

Round 1

Reviewer 1 Report

Establishing an analytical method to investigate the detail of 2D and 3D cell models is an important research, and the spectroscopy is one of the powerful tools to give us a detail molecular information. This manuscript describes silver nanowires with micron meter length for SERS sensing of 2D and 3D cells. In addition, they applied machine learning algorithms for Raman spectra obtained in cells to establish a systematic analysis strategy. Therefore, I believe this manuscript could be published in this journal after they consider the following comments:

1. The authors should introduce the advantages of the proposed method for monitoring living systems compared with other reported methods.

2. How about the reproducibility of different AgNWs?

3. Why Ag ball was generated by electrochemical reduction before rather than after the growth of AgNW? What is the threshold of generation of AgNW from Ag ball?

4. How deep can an AgNW be inserted into the 3D cells with detectable SERS signal?

 Author Response

  1. The authors should introduce the advantages of the proposed method for monitoring living systems compared with other reported methods.

We compared the benefits of SERS over other imaging techniques in the first two sentences of the second paragraph in introduction. In the revised version (second paragraph in the introduction), we clearly pointed out that SERS can provide spectral fingerprints of endogenous biomolecules superior to fluorescence, TEM and SEM.

  1. How about the reproducibility of different AgNWs?

As shown in Figure 2d, 2e and Figure S5, SERS spectra shows reliable reproducibility on different AgNWs. We added corresponding discussion in section 3.2.

  1. Why Ag ball was generated by electrochemical reduction before rather than after the growth of AgNW? What is the threshold of generation of AgNW from Ag ball?

The Ag ball was synthesized to act as a solid connection between the AgNW and the nanopipette. Therefore, we first synthesized the Ag ball on the tip of carbon nanoelectrodes, and then synthesized AgNW on the Ag ball.

Usually, the Ag ball smaller than 2 micrometers does not affect the synthesis of subsequent AgNW. The applied reduction current to synthesize AgNWs was still 2 nA.

We added corresponding discussion in section 3.1.

  1. How deep can an AgNW be inserted into the 3D cells with detectable SERS signal?

Combining the number of acquisition points (shown in Figure S5) and the scanning steps (1 μm) in the SERS measurements, we estimate that the length of the AgNW that can be detected in the 3D cells is 64 ~ 166 μm. We added description of specific scanning step in section 2.2.

Reviewer 2 Report

In the introduction, the authors need to explain why SERS is superior to other imaging techniques. Also compare differences between SERS and Raman scattering.

Line 81: Authors should describe why these two algorithms are preferred and their characteristic features.

Section 2.1 Authors should provide details of the AgNWs synthesis process.

Line 75/76: Extra spacing

Line 129: extra spacing between 37  and oC

For Tables 1 and 2, why authors used tentative assignments? Why these assignments could not be confirmed? Are these assignments already known as referenced in 19-21? If yes, then why did the authors reuse that data?

Author Response

  1. In the introduction, the authors need to explain why SERS is superior to other imaging techniques.

We did compared the benefits of SERS over other imaging techniques in the first two sentences of the second paragraph in introduction. In the revised version, we clearly pointed out that SERS can provide spectral fingerprints of endogenous biomolecules superior to fluorescence, TEM and SEM.

  1. Also compare differences between SERS and Raman scattering.

We added the comparison between SERS and Raman scattering in the third sentence of the second paragraph in introduction.

  1. Line 81: Authors should describe why these two algorithms are preferred and their characteristic features.

To identify the dominant molecular species in the spectra is an unsupervised classification problem, so we chose unsupervised machine learning algorithms to classify the data. Through prior knowledge, we knew that this group of data can be divided into 2 or 3 clusters which means the parameter k of kmeans is determined. In that case, we used the most suitable unsupervised clustering algorithm K-Means++ to classify the data.

In the meanwhile, the importance of Raman shift in different spectrum varies, so we use Random Forest algorithm to filter the unimportant Raman shift to better classify the data.

We added corresponding discussion in line 91-93.

  1. Section 2.1 Authors should provide details of the AgNWs synthesis process.

Detailed AgNWs synthesis process was provided in section 2.1. In the revised version, we also added discussion on the effect of capping agent in section 3.1.

  1. Line 75/76: Extra spacing

We have deleted the extra spacing.

  1. Line 129: extra spacing between 37  and oC

We have deleted the extra spacing.

  1. For Tables 1 and 2, why authors used tentative assignments? Why these assignments could not be confirmed? Are these assignments already known as referenced in 19-21? If yes, then why did the authors reuse that data?

The Raman spectrum of any given substance is interpreted by the use of those known group frequencies and thus it is possible to characterize the substance as one containing a given type of group or groups. However, it is also well known that interference or perturbation may cause a shift of the characteristic bands due to (a) the electronegativity of neighboring groups or atoms, or (b) the spatial geometry of the molecule.[1] Therefore, for the sake of rigorous expression, some paper use “tentative assignments” to show that the assignment is just according to the references without further investigation.[2] For the same reason, we choose using “tentative assignment” here.

[1] George Socrates, 2004, Infrared and Raman Characteristic Group Frequencies, Wiley.

[2] Živanović, V.; Semini, G.; Laue, M.; Drescher, D.; Aebischer, T.; Kneipp, J., Chemical mapping of leishmania infection in live cells by SERS microscopy. Anal. Chem. 2018, 90 (13), 8154-8161.

Reviewer 3 Report

In this manuscript, the authors fabricated an ultra-long one-dimensional silver nanowire as a SERS probe in order to acquire interior information from MCF-7 breast cancer cells in 3D cell culture model. With the aid of algorithms, such as PCA and a machine learning method, nucleic acid-rich and protein-rich regions can be significantly distinguished in the 2D and 3D models. I feel the manuscript meets the scope of Biosensors, and below are some comments.

Suggestions to authors for improving the manuscript:
1. What’s the resonance frequency/wavelength of the ultra-long SERS nanowire?

2. Could the conclusion be addressed based on regular spectrum analysis (like peak ratio, presence/absence of peaks) rather than PCA or KNN models?

3. What are the limitations/drawbacks of the KNN algorithm (random forest and k-means)?

Author Response

  1. What’s the resonance frequency/wavelength of the ultra-long SERS nanowire?

The resonance wavelength of the ultra-long AgNW is around 633 nm. Considering that measuring the extinction spectrum of a single AgNW is very difficult and currently impossible, we tested the resonance properties of AgNW using three wavelengths of laser light. The laser of 633 nm produces a stronger plasmonic resonance on the AgNW surface than the lasers of 488 nm and 785 nm. We added corresponding discussion in section 2.2.

  1. Could the conclusion be addressed based on regular spectrum analysis (like peak ratio, presence/absence of peaks) rather than PCA or KNN models?

As shown in Figure 2d~2f, the Raman spectra of the 2D and 3D cell models demonstrated remarkable similarity in several strong bands as expected, because that both models were formed by MCF-7 cells. Therefore, it is hard to analyze the spectra in depth by regular spectrum analysis (like peak ratio, presence/absence of peaks) and we must introduce computational way to capture key features and identify biomolecules in spatial resolution. We added corresponding discussion in section 3.2.

  1. What are the limitations/drawbacks of the KNN algorithm (random forest and k-means)?

KNN is a simple classification algorithm with wide application and little influence from abnormal data. However, the computational cast will be high if there is a large amount of data. Besides, when the data amounts of different classes are not balanced, the prediction accuracy of the classes with small data amounts would be low. Fortunately, neither is present in our system.

K-Means++ is an unsupervised clustering algorithm that can cluster unlabeled data. The limitations of K-Means++ algorithm may be the requirement of determining the number of clusters in advance. Meanwhile, the result of classification depends heavily on the initialization of the cluster center. The result is not necessarily global optimal, but only local optimal. However, K-Means++ is still our best choice for its simple principle, convenient implementation, fast convergence and better clustering accuracy.

Random Forests is a supervised algorithm with very high accuracy and efficiency on large data set. It is able to process high dimensional data without dimension reduction. However, when the number of decision trees is too large, the space and time required for training will be large, which will lead to slower inference time. Therefore, in practical application, if the real-time requirement is very high, it is better to choose other algorithms with fast inference time.

We added corresponding discussion about their characteristic features in line 91-93.

Round 2

Reviewer 2 Report

No comments.